# The Effectiveness of Anti-Apoptotic Agents to Preserve Primordial Follicles and Prevent Tissue Damage during Ovarian Tissue Cryopreservation and Xenotransplantation

**DOI:** 10.3390/ijms22052534

**Published:** 2021-03-03

**Authors:** Sanghoon Lee, Hyun-Woong Cho, Boram Kim, Jae Kwan Lee, Tak Kim

**Affiliations:** Department of Obstetrics and Gynecology, Korea University College of Medicine 73, Inchon-ro, Seongbuk-gu, Seoul 02841, Korea; mdleesh@gmail.com (S.L.); limpcho82@korea.ac.kr (H.-W.C.); kbr12190409@naver.com (B.K.); tkim@korea.ac.kr (T.K.)

**Keywords:** fertility preservation, apoptosis, ovarian tissue transplantation, Z-VAD-FMK, sphingosine-1-phosphate

## Abstract

The purpose of this study is to investigate the effectiveness of sphingosine-1-phosphate (S1P) and Z-VAD-FMK (Z-VAD) as anti-apoptotic agents to preserve ovarian function and prevent tissue damage during ovarian tissue cryopreservation and transplantation. This study consisted of two steps, in vitro and in vivo. In the first step, human ovarian tissues were cryopreserved using slow-freezing media alone, S1P, or Z-VAD (control, S1P, Z-VAD group); based on the outcomes in these groups, Z-VAD was selected for subsequent xenotransplantation. In the second step, human frozen/thawed ovarian tissues were grafted into fifty mice divided into three groups: slow-freezing/thawing and transplantation without an anti-apoptotic agent (Trans-control) and xenotransplantation with or without Z-VAD injection (Trans-Z-VAD-positive and Trams-Z-VAD-negative groups, respectively). In the first step, the Z-VAD group had a significantly higher primordial follicular count than the S1P (*p* = 0.005) and control groups (*p* = 0.04). Transplanted ovarian tissues were obtained 4 weeks after transplantation (second step). Angiogenesis was significantly increased in the Z-VAD-negative (*p* = 0.03) and -positive (*p* = 0.04) groups compared to the control group. This study demonstrated that slow-freezing and transplantation with Z-VAD is an effective method for preserving primordial follicle counts, decreasing double-strand DNA breaks, and increasing angiogenesis in a mouse model. Further molecular and clinical studies are needed to confirm these results.

## 1. Introduction

Primary ovarian insufficiency (POI) is a major concern in reproductive-age women with cancer. Radical surgery, chemotherapy, and/or radiation may result in loss of ovarian function [1]. Although standard fertility preservation options for women include embryo or oocyte preservation, these techniques require ovarian stimulation for approximately 2 weeks, which may result in delayed cancer treatment [2].

According to the Practice Committee of the American Society of Reproductive Medicine (ASRM), ovarian tissue cryopreservation is considered as an acceptable method for preserving female fertility when gonadotoxic treatments cannot be delayed or in patients before puberty or when there is desire to cryopreserve more than just a few oocytes [3]. It is the best and only choice for preserving fertility for children, adolescents, and young adult cancer patients who need urgent chemotherapy and do not have enough time to induce ovulation [4,5]. This method does not require ovarian stimulation or a sperm donor. In addition, the hormonal function of the ovary can be restored, improving the quality of life of young women [6,7,8].

Successful ovarian tissue cryopreservation and transplantation techniques are highly dependent on angiogenesis [9]. Since new blood vessel formation and maturation takes up to 10 days after transplantation, two-thirds of ovarian tissue may be lost to initial ischemia [10,11]. Therefore, it is important to improve the survival rate of ovarian tissue during cryopreservation and transplantation in order to prolong the life of the transplanted tissue. Many approaches have been investigated to minimize ischemic injury and improve angiogenesis in ovarian tissues. It has been reported that anti-apoptotic agents, such as sphingosine-1-phosphate (S1P) and Z-VAD-FMK (Z-VAD), may enhance the success rate of ovarian tissue transplantation [12,13,14,15]. However, there is a lack of data on the effects of S1P and Z-VAD on follicle preservation and the success of ovarian tissue transplantation. There have been no studies comparing the effects between anti-apoptotic agents including S1P and Z-VAD. In addition, optimal doses or method of anti-apoptotic agent administration during cryopreservation and transplantation process of ovarian tissue have not been standardized.

The purpose of this study was to investigate the effectiveness of S1P and Z-VAD as anti-apoptotic agents to preserve primordial follicles as well as prevent tissue damage and angiogenesis after ovarian tissue cryopreservation and transplantation.

## 2. Results

This study consisted of two steps, in vitro and in vivo (Figure 1). In the first step, 171 pieces of ovarian tissue were evenly distributed into the control (without an anti-apoptotic agent), S1P (S1P administration before freezing), and Z-VAD (Z-VAD administration before freezing) groups. In the second step, 191 pieces of ovarian tissue were also distributed into the Trans-control (without an anti-apoptotic agent), Trans-Z-VAD-negative (Z-VAD administration before freezing), and Trans-Z-VAD-positive (Z-VAD administration before freezing and Z-VAD injection into transplanted mouse) groups.

### 2.1. Histologic Evaluation and Primordial Follicular Counts

After the freezing–thawing process, the follicle density of the Z-VAD group was significantly higher than that of the control (11.23 ± 2.48 vs. 7.05 ± 1.52 count/mm^2^, *p* = 0.04, Student’s t-test) and S1P groups (11.23 ± 2.48 vs. 3.22 ± 1.18 count/mm^2^, *p* < 0.005, Student’s t-test) (Figure 2). However, there was no significant difference in follicle density between the Trans-Control (3.27 ± 1.52 count/mm^2^), Trans-Z-VAD-negative (3.40 ± 0.91 count/mm^2^), and Trans-Z-VAD-positive (3.71 ± 1.45 count/mm^2^) groups after xenotransplantation (Figure 2).

### 2.2. Follicular Cell Proliferation, Double-Strand DNA Damage, and Angiogenesis

Figure 3 displays representative images of follicular cell proliferation and double-strand DNA damage in the frozen–thawed tissues determined by IHC. In addition, representative images of follicular cell proliferation, double-strand DNA damage, and angiogenesis of transplanted ovarian tissues are presented in Figure 4. In the Z-VAD group, the proportion of primordial follicles expressing Ki-67 was higher than that in the S1P (59.45 ± 17.21% vs. 27.37 ± 8.94%, *p* = 0.003, Student’s t-test) and control (59.45 ± 17.21% vs. 30.40 ± 11.30%, *p* = 0.004, Student’s t-test) groups (Figure 3). Consistently, the proportion of Ki-67-positive primordial follicles was higher in Trans-Z-VAD-positive groups than in the control group (58.15 ± 12.96% vs. 22.02 ± 12.33%, *p* < 0.05, Student’s t-test) (Figure 4).

A significantly lower percentage of gamma H2A histone family member X (γH2AX)-positive primordial follicles were observed in the Z-VAD group compared to the S1P group (24.28 ± 7.20% vs. 46.12 ± 9.38%, *p* = 0.03, Mann–Whitney U test) (Figure 3). Injection of xenograft mice with Z-VAD (Trans-Z-VAD-positive group) resulted in a significantly lower percentage of γH2AX-positive primordial follicles than that in the Trans-Z-VAD-negative group (33.14 ± 6.95% vs. 45.23 ± 4.78%, *p* = 0.04, Mann–Whitney U test) (Figure 4). In addition, angiogenesis (shown by the CD31-positive) was more prominent in the Trans-Z-VAD-negative (1.63 ± 0.14 vs. 1.25 ± 0.12 vessels/mm^2^, *p* = 0.03, Mann–Whitney U test) and -positive groups (1.61 ± 0.11 vs. 1.25 ± 0.12 vessels/mm^2^, *p* = 0.04, Mann–Whitney U test) compared to the control (Figure 4).

### 2.3. Western Blot

Anti-Müllerian hormone (AMH) (60 kDa) expression was evaluated by Western blot, revealing that treatment with Z-VAD (3.89 ± 1.19 AMH/actin relative intensity) and S1P (3.72 ± 1.23 AMH/actin relative intensity) did not significantly increase the AMH production level compared to the control group (2.39 ± 0.75 AMH/actin relative intensity) in frozen–thawed human ovarian tissues (Figure 5). Consistently, Z-VAD injection did not significantly improve AMH production in the xenotransplant model (Trans-Control: 2.2 ± 0.8, Trans-Z-VAD-negative: 3.2 ± 0.9, Trans-Z-VAD-positive: 3.9 ± 1.0 AMH/actin relative intensity) (Figure 5).

## 3. Discussion

Despite recent improvements in ovarian tissue cryopreservation and transplantation techniques, ischemia during the initial days after ovarian transplantation is a major obstacle to success. In addition, follicular apoptosis can occur in frozen/thawed ovarian tissue [12,16,17,18]. Several studies demonstrated that apoptotic follicles increased shortly after ovarian tissue transplantation [19,20,21]. Therefore, revascularization and prevention of apoptosis are key factors in ensuring ovarian tissue graft survival.

S1P is largely studied for its anti-apoptotic and pro-angiogenic properties. S1P, which is synthesized by sphingosine kinase from sphingosine, regulates a variety of proliferative cellular processes including cell growth and differentiation, and it counterbalances the apoptosis [22,23]. S1P and ceramides play an important role in apoptosis. S1P antagonizes high ceramide levels, which induces apoptosis [13]. Especially for oocyte, the S1P-related pathway has been identified as a crucial mediator regulating apoptosis [24]. Moreover, recent studies suggested that S1P is a modulator of angiogenesis [25]. In previous studies, S1P protected vitrified ovarian grafts from ischemic reperfusion injury and promoted neo-angiogenesis in ovarian transplants [12,13]. However, there is a disadvantage that it would require continuous administration (pump) or injections direct into the ovaries due to the short plasma half-life of S1P [26].

Another alternative, Z-VAD, is a cell permeable synthetic broad-spectrum caspase inhibitor. The caspase family of cysteine proteases presents the main effector molecules of apoptosis [27]. Since caspases trigger cells to execute apoptosis by cleaving and altering the functions of diverse intracellular proteins, caspase inhibition could be an attractive strategy to prevent follicular apoptosis. Supplementation with a global pan-caspase inhibitor (Z-VAD) could help to prevent follicular apoptosis [28]. Although there are few studies of Z-VAD in a xenograft model of cryopreserved ovarian tissue, it prevented apoptosis and ischemic/reperfusion damage in several tissues, such as the myocardium, brain, liver, muscle, small intestine, and pancreas upon exposure to ischemia [12,15,29].

This study showed that the administration of Z-VAD improved follicle preservation and follicular cell proliferation and prevented DNA damage in follicles during the freezing–thawing process. In the transplantation process, Z-VAD injection enhanced angiogenesis and the follicular proliferation of ovarian transplants and reduced DNA damage. However, S1P did not improve the preservation and proliferation of follicles and DNA damage during the freezing–thawing process.

Consistent with the present findings, previous studies have shown that Z-VAD inhibits follicle apoptosis during cryopreservation or in vitro and protects against ovarian tissue damage after transplantation. Z-VAD reduced cryopreservation-induced apoptosis [28], maintained the metabolic activity of granulosa cells, and prevented granulosa cell death [15]. In addition, the in situ administration of Z-VAD to ovarian tissues before transplantation might help preserve follicles and prevent apoptosis after transplantation [15].

However, S1P treatment did not prevent follicle loss when compared to the control group, and the follicle counts and DNA breakage in the S1P group were inferior to those in the Z-VAD group, which was inconsistent with previous studies. S1P showed a protective effect in the case of ovarian follicles both in vitro and in mouse models [14,30]. S1P prevented follicle loss and preserved the ovarian reservoir after transplantation of vitrified ovarian grafts using a mouse model [13].

There are two possible hypotheses to explain the differences from previous studies. First, S1P was shown to have a paradoxical toxic effect, as microinjecting keratinocytes with S1P induced strong cell growth arrest and reduced cell proliferation in vitro [31]. Secondly, S1P may have had little effect on the freezing–thawing process. S1P has been shown to be more effective at suppressing apoptosis during tissue transplantation than cryopreservation [13].

There are some limitations in our study. First, a heterotopic subcutaneous model was used in this study. The microenvironments of transplanted ovarian cortex might differ slightly from their original position in patients [9]. However, the subcutaneous model showed a higher follicle survival rate compared to the sub-renal model. Another report found no significant differences in endocrine function between a subcutaneous model and transplantation methods [9]. Second, this study did not include S1P in the in vivo study, because S1P administration did not prevent primordial follicle loss in vitro in the first place. In addition, since S1P has a shorter half-life than Z-VAD, a continuous pump will be required in vivo for S1P, which may lead to biases between the two anti-apoptotic agents. Moreover, when the catheter material is used for continuous pump, it may cause complications such as skin irritation and inflammation [32]. Third, since the results of this study were obtained from analysis of tissues harvested 4 weeks after tissue transplantation, there are limitations in assessing the long-term effects. However, neo-angiogenesis and follicular survival following ischemic damage occur approximately 5 days after xenotransplantation [11]. Therefore, 4 weeks may be sufficient to assess the effect on the tissue. Fourth, we did not assess hormone levels in the blood, including AMH, follicular-stimulating hormone (FSH) and luteinizing hormone (LH), although AMH expression was confirmed by Western blot analysis. In previous human study, endocrine function recovery after ovarian transplantation was shown [33]. Although the tendency to increase AMH expression in Z-VAD group was observed in this study (Figure 5), there was no statistical significance between groups. Assessment on the blood level of AMH might be needed for evaluation of endocrine functions. further studies should investigate the effects of anti-apoptotic agents on ovarian transplantation outcomes in terms of endocrine function. Finally, the effects of ovarian tissue cryopreservation and transplantation should ultimately be assessed with pregnancy and live birth. Therefore, further clinical trials are needed.

However, this study has several strengths. To our knowledge, this is not only the first study comparing the effects of the anti-apoptotic agents, S1P and Z-VAD, but also novel in showing the effects of Z-VAD when it is administered before freezing and after transplantation.

In conclusion, this study suggested that slow-freezing and transplantation with Z-VAD could be considered an effective method for preserving fertility with respect to primordial follicle counts, decreased double-strand breaks, and increased angiogenesis using a mouse model. This is the first study to examine the effects of Z-VAD when it is administered before freezing and after transplantation. Further molecular and clinical studies are needed to confirm the results obtained.

## 4. Materials and Methods

### 4.1. Study Design

Human ovarian cortex tissues were obtained from 13 patients (16–33 years old) that underwent benign ovarian surgery. Informed consent was obtained from all patients preoperatively or, if patients were under 18, from a parent and/or legal guardian. Ovarian tissues from patients were punctured using a biopsy punch (Kai Industries Co., Ltd., Seki City, Japan) to generate identical biopsies 4 mm in diameter and 1 mm thick [34]. Between 20 and 84 pieces of ovarian tissue were obtained per patient, and these pieces from per patient were all evenly assigned to each group.

In the first step of this study, 171 pieces of ovarian tissue were evenly distributed into the control, S1P, and Z-VAD groups. These pieces of ovarian tissue were assessed for primordial follicle density, follicular cell proliferation, and double-strand DNA damage after the freezing–thawing process. In the second step, 191 pieces of ovarian tissue were also distributed into the Trans-control, Trans-Z-VAD-negative, and Trans-Z-VAD-positive groups. These ovarian tissue samples were frozen–thawed and then transplanted into ovariectomized severe combined immunodeficient (SCID) mice [34]. Four weeks after transplantation, the grafts were harvested from sacrificed mice.

Compared to S1P, Z-VAD showed better results regarding preantral follicles, double-strand DNA breaks, and cell proliferation in vitro after the freezing–thawing process; thus, Z-VAD was used in xenotransplantation.

### 4.2. Anti-Apoptotic Agents

S1P (Sigma-Aldrich, St. Louis, MO, USA) and Z-VAD (R&D system, Minneapolis, MN, USA) were purchased, prepared as described by the manufacturer, and used at concentrations that were previously shown to be effective on human ovarian tissue (200 and 50 μM, respectively) [15,30].

### 4.3. Slow-Freezing Protocol

The slow-freeze protocol has been described previously [35,36]. Ovarian tissue fragments were transferred to a basic solution containing 20% human serum albumin (HSA; Green Cross, Gyeonggi-do, Korea) in M199 culture medium (catalogue #M4530; Sigma-Aldrich). Cryoprotectants were added by serial dilution in three steps. Tissue fragments embedded in 20% HSA-supplemented M199 culture medium were exposed to 7.5% dimethyl sulfoxide (DMSO; catalogue #D2650, Sigma-Aldrich) for 5 min at 4 °C, followed by 10% and 12.5% DMSO for 15 min each. All procedures were carried out on ice. Ovarian tissue fragments were transferred to individual freezing tubes containing 1 mL of medium. S1P or Z-VAD was added to the medium at appropriate amounts to obtain the final concentration (200 and 50 μM, respectively). Thus, S1P or Z-VAD still remains in the cryo-tube during the thawing process. The cryotubes were cooled in a programmable controlled-rate freezing device (Planer PLC, Middlesex, UK) with a slow-freeze protocol, as described previously, which featured cooling from 4 to −7 °C at a rate of −2.0 °C/min, followed by manual seeding and cooling to −40.0 °C at a rate of −0.3 °C/min and then to −140.0 °C at a rate of −10 °C/min. Finally, the samples were stored at −196.0 °C in liquid nitrogen.

### 4.4. Slow Freezing–Thawing Protocol

Tissue thawing was performed as described previously [35]. Stored cryotube vials were transferred from liquid nitrogen to shaking baths at 37 °C. After the vial was thawed completely, half of the supernatant was removed and replaced with an equal volume of wash solution containing 5% DMSO in basic medium. The vial was incubated at 20–25 ºC for 10 min. After incubation, half of the supernatant was removed again and replaced with an equal volume of wash solution. The vial was incubated at room temperature for 5 min.

### 4.5. Xenotransplantation into SCID Mice

Thawed human ovarian tissue was washed with phosphate-buffered saline (PBS) and stored in normal saline. Transplantation was also performed as described previously [34]. Fifty female C.B-17/Fox Chase SCID mice (Trans-control: *n* = 10, Trans-Z-VAD-negative: *n* = 20, Trans-Z-VAD-positive: *n* = 20), 6 weeks of age, were ovariectomized before xenotransplantation under anesthesia via a small incision in the body wall that was sutured with 6-0 silk thread [34]. Frozen/thawed human ovarian tissue was transplanted into the back muscles of mice (4 pieces per mouse). Z-VAD was administered daily by intraperitoneal injection at doses of 2 mg/kg for 7 consecutive days [37].

### 4.6. Histologic Evaluation

Four weeks after transplantation, the grafts were retrieved from the transplanted mice. Ovarian tissue samples from each of the three groups were fixed with 4% formaldehyde for histological assessment. After fixation, tissues were washed three times with PBS and dehydrated in stages with graded ethanol solutions (50%, 70%, 80%, 90%, and 100%). After routine paraffin embedding, samples were cut (2 μm thickness), stained with hematoxylin and eosin, and then examined using a microscope to determine follicle number and density. Three researchers independently examined the samples repeatedly. To avoid repetition, only follicles containing oocytes with visible nuclei were counted. Morphologically intact primordial follicles were identified based on oocyte integrity by skilled pathologists. A primordial follicle was defined as an oocyte surrounded by a single fusiform granule cell [34]. The follicular densities were estimated by evaluation of follicles per field of view of ovarian tissue [30].

### 4.7. Immunohistochemistry (IHC) Evaluation

Immunohistochemistry (IHC) staining with γH2AX, Ki-67, and CD31 was performed to evaluate the strength of DNA damage, cell proliferation, and angiogenesis in cryopreserved and xenografted human ovarian tissue. Twenty samples in each of the three groups were evaluated by gamma-h2ax, Ki-67, and CD31 staining. For IHC staining, unstained paraffin-embedded ovarian tissue slides were deparaffinized with xylene and rehydrated with a series of ethanol solutions (50%, 70%, 80%, 90%, and 100%).

The slides were incubated in 1× citrate buffer (pH 6.0; C9999, Sigma-Aldrich, St. Louis, MO, USA) for 15 min in a microwave for heat-induced epitope antigenic retrieval. The samples were blocked with peroxide blocking solution for 10 min and washed with washing buffer (1× Tris-Tween-20 buffer in deionized water). After blocking, sections were incubated with antibodies to gamma-h2ax (1:250 dilution, IHC-00059, Bethyl Laboratories, Montgomery, TX, USA), Ki-67 (1:100 dilution, 275R-14, Cell Marque, Rocklin, CA, USA), and CD31 (1:200 dilution, Abcam, UK) at 4 °C overnight.

Following incubation, the slides were washed three times and incubated for 1 h with Polink-2 HRP plus rabbit antibodies (D41-18, GBI LABS, Bothell, USA). After washing, the sections were treated with EnVision+ horseradish peroxidase and liquid 3,30-diaminobenzidine tetrahydrochloride (DAB)+ substrate and counterstained with Mayer’s hematoxylin (Scytek, Logan, UT, USA).

Slides were dewatered and mounted before microscopic examination. The number of primordial follicles stained with Ki-67 and CD31-positive vessels was counted in all visual microscopic fields per sample and were expressed as mean per sample [38].

### 4.8. Western Blot

Western blot analyses for anti-Müllerian hormone (AMH) production were performed on human ovarian tissues. Western blot analyses were performed as previously described [34]. Proteins were detected by incubation with anti-AMH (1:100 dilution, catalogue #ab103233; Abcam), and anti-β-actin (1:1000, catalogue #sc-47778; Santa Cruz Biotechnology, Santa Cruz, CA, USA) primary antibodies at 4 °C overnight with gentle shaking. Following incubation, the proteins incubated with goat anti-rabbit secondary antibody (1:5000 dilution, catalogue #ab6721; Abcam) in 1× tris-buffered saline (TBS) containing 3% skim milk and 0.05% Tween-20 at room temperature for 90 min. Immunoreactive proteins were visualized by chemiluminescence using Clarity Western ECL Substrate (catalogue #1705060, Bio-Rad Laboratories) and detected on medical X-ray blue film (Agfa-Gevaert, Mortsel, Belgium). Western blotting was repeated five times for AMH analyses. The band intensities were quantified relative to intensity of beta-actin by densitometry using imageJ software.

### 4.9. Statistical Analysis

All the statistical analyses were performed using SPSS version 12.0 software (SPSS Inc., Chicago, IL, USA). The results of follicle counts, number of Ki-67- and γH2AX-positive cells, proportion of the CD31-positive area, and AMH expression within the samples were compared. When the basic data or transformed data were normally distributed by the Shapiro–Wilk normality test, Student’s t-test was used for comparisons. When the basic data and transformed data were not distributed normally, we used the Mann–Whitney U-test for comparisons. A *p*-value < 0.05 was considered statistically significant.

## Figures and Tables

**Figure 1 ijms-22-02534-f001:**
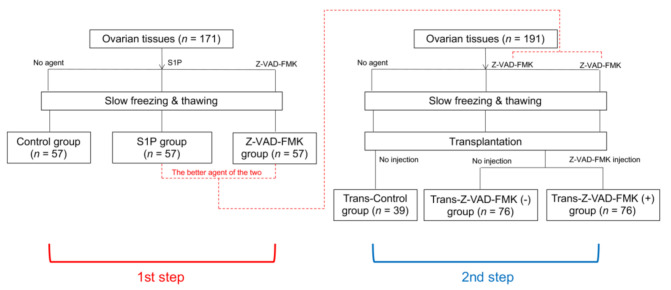
Study scheme.

**Figure 2 ijms-22-02534-f002:**
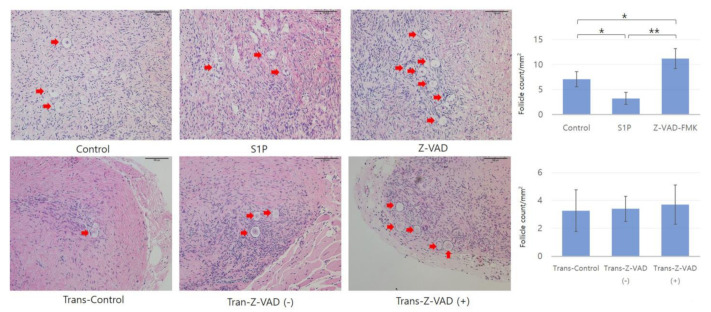
Histopathological findings and primordial follicle density in frozen–thawed human ovarian tissues and human ovarian tissues after xenotransplantation (magnification × 200). Student’s independent t-test showed that the follicular density was significantly different among the three groups (each group, *n* = 20, * *p* < 0.05, ** *p* < 0.005)) in frozen–thawed human ovarian tissue. However, there was no significant difference in follicle density among three groups (control: *n* = 10, trans-Z-VAD (-) and (+): *n* = 20) in human ovarian tissue after transplantation. Red arrows indicate primordial follicles. * *p* < 0.05.

**Figure 3 ijms-22-02534-f003:**
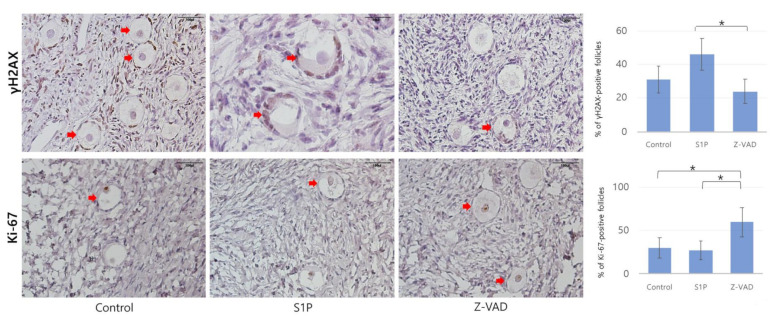
Histological features of frozen–thawed human ovarian tissues in the three groups evaluated with immunohistochemical staining for Ki-67 (magnification × 200, control: *n* = 19, S1P: *n* = 18, Z-VAD: *n* = 19, * *p* < 0.05) and γH2AX (control and Z-VAD: magnification × 200, S1P: magnification × 400) (control: *n* = 18, S1P: *n* = 19, Z-VAD: *n* = 18, * *p* < 0.05). Red arrows indicate Ki-67-positive or gamma H2A histone family member X (γH2AX)-positive primordial follicles. S1P: sphingosine-1-phosphate, Z-VAD: Z-VAD-FMK.

**Figure 4 ijms-22-02534-f004:**
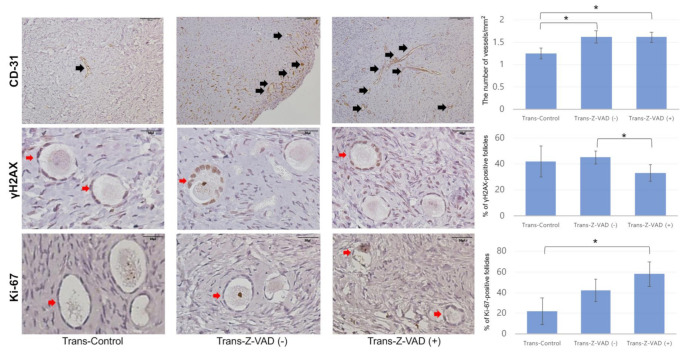
Histological features of human ovarian tissues after xenotransplantation in the three groups evaluated with immunohistochemical staining for CD-31 (magnification × 400, control: *n* = 10, Trans-Z-VAD (-): *n* = 19, Trans-Z-VAD ( + ): *n* = 20, * *p* < 0.05), Ki-67 (magnification × 400, control: *n* = 9, Trans-Z-VAD (-): *n* = 19, Trans-Z-VAD ( + ): *n* = 18, * *p* < 0.05), and γH2AX (magnification × 400, control: *n* = 10, trans-Z-VAD (-): *n* = 18, trans-Z-VAD ( + ): *n* = 18, * *p* < 0.05). Black arrows indicate CD-31-stained cells. Red arrows indicate Ki-67/γH2AX-positive primordial follicles.

**Figure 5 ijms-22-02534-f005:**
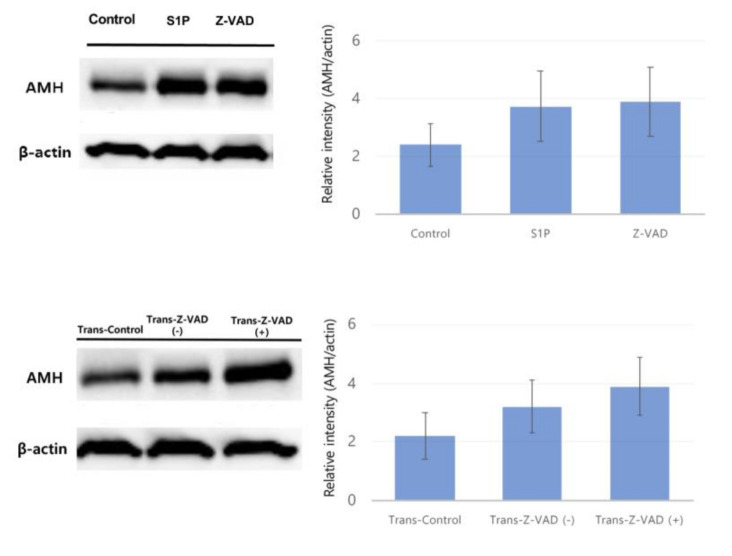
Western blot analysis to evaluate cell apoptosis based on anti-Müllerian hormone production in frozen–thawed human ovarian tissues and human ovarian tissues after xenotransplantation. Cropped blots are presented.

## Data Availability

The datasets generated and/or analyzed during the current study are available from the corresponding author on reasonable request.

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
