# Peer review of "The Effectiveness of Anti-Apoptotic Agents to Preserve Primordial Follicles and Prevent Tissue Damage during Ovarian Tissue Cryopreservation and Xenotransplantation"

_ijms, 2021, doi:10.3390/ijms22052534_

Round 1

Reviewer 1 Report

The title:  Impact of Anti-Apoptotic Agents on Ovarian Tissue Cryopreservation and Xenotransplantation for Fertility Preservation  does not describe the manuscript, it should be more specific

The abstract reflects the contents of the manuscript and is well structured summarizing the content of the paper.

The introduction is clear but the purpose of the work should be more clearly described.

The methods section is clear and without methodological failures.

The result section needs clarification as to the red arrows in figure 3 seems to be misplaced. Also, AMH quantification should be correlated to follicular density and not only compared with actin quantification.

The discussion is correct and limitations clearly stated and the conclusions is relevant .

Author Response

The title:  Impact of Anti-Apoptotic Agents on Ovarian Tissue Cryopreservation and Xenotransplantation for Fertility Preservation  does not describe the manuscript, it should be more specific

Answer: We thank the reviewer for their comments, and we changed the title as you suggested. 

The abstract reflects the contents of the manuscript and is well structured summarizing the content of the paper.

The introduction is clear but the purpose of the work should be more clearly described.

Answer: We thank the reviewer for their comments, and we modified the purpose in introduction of manuscript as you suggested (page 2, line 63-66). 

The methods section is clear and without methodological failures.

The result section needs clarification as to the red arrows in figure 3 seems to be misplaced. Also, AMH quantification should be correlated to follicular density and not only compared with actin quantification.

Answer: We thank the reviewer for their comments, and we revised the figure 3 as you suggested. Regarding to AMH quantification, we estimated the density of primordial follicles which is not positively related to AMH production. AMH is produced by granulosa cell of preantral or antral follicles. Since AMH is not produced by primordial follicles, we assumed that AMH quantification with actin is better than correlation to follicular density.

The discussion is correct and limitations clearly stated and the conclusions is relevant 

Reviewer 2 Report

Dear authors,

the manuscript of Lee et al. deals with the important question of how to improve the angiogenesis and lowering the apaoptosis in transplanted ovarian grafts. The authors present in vitro and in vivo animal studies. The manuscript is well written but I have some issues with the experimental design and performance and the possible transfer to the clinic in human:

  • how did you solve S1P and Z-Vad, because both are soluble in alcohol or DMSO, respectively?
  • did you use a vehicle control for freezing and thawing with the solvent alone?
  • did you count the follicle density in fresh tissue without freezing /thawing as a control?
  • primary follicles are not always distrubuted regularly in the ovarian cortex; please explain the counting of 1 and 4 respectively microscopic fields, because this is dependent on the magnification and does not represent the distribution in the whole biopsy
  • what does FMK mean in figure 1?
  • how do you explain the reduction in follicle density in the grafts in all groups?
  • why did you choose CD31 as a marker for angiogenesis and not VEGF?
  • data about endocrine function of the transplanted tissues would improve your data
  • which size did you detect for caspase 3? Please check if you detected total caspase 3 or cleaved caspase 3?
  • in my opinion the injection of Z-VAD would not be possible after transplantation in human, unfortunately; therefore I suggest rethinking the design according to clinical routine in human for further studies
  • please indicate more details about the preparation of the ovarian cortex; in some studies strips are transplanted instead of biopsies and the number of remaining fibroblasts influences the angiogenesis a lot
  • clinical data in human reassume the hormonal activity of the grafts, that are transplanted in a peritoneal pocket in most cases and orientated towards the fallopian tube, after 4 to 8 weeks; therefore the hormone profile would also be beneficial for this manuscript
  • reference 35 is named for the slow-freezing and thawing protocol, but the table in the refrence differs from the design in the manuscript; please check and declare
  • do you know if S1P and Z-VAD are stable during freezing and thawing?
  • especially for the thawing, the exposure time of the tissue in DMSO at room temperature should be as short as possible in order to avoid cellular damage. DMSO and warm temperature harm the tissue; please explain

Looking forward for the revised version, good luck

  •  

Author Response

Reviewer 2

Dear authors,

the manuscript of Lee et al. deals with the important question of how to improve the angiogenesis and lowering the apaoptosis in transplanted ovarian grafts. The authors present in vitro and in vivo animal studies. The manuscript is well written but I have some issues with the experimental design and performance and the possible transfer to the clinic in human:

how did you solve S1P and Z-Vad, because both are soluble in alcohol or DMSO, respectively?

Answer: We thank the reviewer for their comments. In this study, S1P and Z-VAD were dissolved in methanol and DMSO, respectively.

did you use a vehicle control for freezing and thawing with the solvent alone?

Answer: We thank the reviewer for their comments. In control group, DMSO was used during the slow freezing process but methanol was not used.

did you count the follicle density in fresh tissue without freezing /thawing as a control?

Answer: We thank the reviewer for their comments. As the reviewer commented, it would be better if we performed follicle count on fresh tissue. However, it is a well-known that the follicle count decreases due to cryodamage, apoptosis, or ischemic injury in cryopreservation and transplantation. In addition, the purpose of this study is to find out how to reduce follicle loss and tissue damage during cryopreservation and transplantation by apoptosis. Therefore, we used frozen-thawed ovarian tissue without apoptotic agent as control like other previous studies.

primary follicles are not always distrubuted regularly in the ovarian cortex; please explain the counting of 1 and 4 respectively microscopic fields, because this is dependent on the magnification and does not represent the distribution in the whole biopsy

Answer: We thank the reviewer for their comments. In this study, follicle count was performed in all microscopic field. Because the pieces of ovarian tissue are very small (4mm diameter), each of them was able to cover with 4 microscopic fields, but the manuscript (page 11, line 324-326) was revised to avoid misunderstanding; “The number of primordial follicles stained with Ki-67 and CD-31-positive vessels was counted in all visual microscopic fields per sample ~”.

what does FMK mean in figure 1?

Answer: We thank the reviewer for their comments. The full name of the anti-apoptotic agent we used in our study is Z-VAD-FMK and Z-VAD is the abbreviation. Figure 1 has been modified to prevent misunderstanding.

how do you explain the reduction in follicle density in the grafts in all groups?

Answer: We thank the reviewer for their comments. Successful ovarian tissue cryopreservation and transplantation techniques are highly dependent on angiogenesis [1]. In general, new blood vessel formation and maturation take up to 10 days after transplantation. Therefore, about two-thirds of ovarian tissue may be lost due to initial ischemia [2, 3].

why did you choose CD31 as a marker for angiogenesis and not VEGF?

Answer: We thank the reviewer for their comments. Both CD-31 and VEGF are well defined markers of angiogenesis. CD31 is highly expressed on the surface of endothelial cells and well established for the monitoring of vessel density in tissue. In addition, CD31 is used to monitor of the vessel formation after ovarian tissue transplantation in many studies [4-8].

data about endocrine function of the transplanted tissues would improve your data

Answer: We thank the reviewer for their comments and absolutely agree your opinion. In this study, we tried to assess the blood level of E2, FSH, and AMH. However, the maximum amount of whole blood that can be extracted in the laboratory from a mouse (20g) is 1.0-1.5ml and the amount of serum is usually less than 0.5-1ml. Therefore, only one hormone can be measured in this condition. In addition, when we compared the results of AMH test between commercial ELIZA kit and blood test of hospital laboratory, we failed to show reliable and consistent results. For these reasons, AMH was measured using western blot from ovarian tissue. We described this point as limitation of our study in discussion section (page 9, line 216-218).

which size did you detect for caspase 3? Please check if you detected total caspase 3 or cleaved caspase 3?

Answer: We thank the reviewer for their comments. Actually, the data that we showed is the detection for total caspase 3. Although we tried to detect cleaved caspase-3, we failed to detect this for unknown reason. Therefore, we deleted the results on caspase-3 in manuscript and revised the figure 5. We apologize for the confusion.

in my opinion the injection of Z-VAD would not be possible after transplantation in human, unfortunately; therefore I suggest rethinking the design according to clinical routine in human for further studies

Answer: We thank the reviewer for their comments. As you commented, currently, there is no clinical trial which used Z-VAD-FMK, but it is expected to be potentially used in human studies. Z-VAD-FMK could prevent follicular apoptosis in a xenograft model of cryopreserved ovarian tissue [9], It also prevented apoptosis and ischemic/reperfusion damage in several tissues, such as the myocardium, brain, liver, muscle, small intestine, and pancreas upon exposure to ischemia [10-12]. Therefore, we would appreciate if you could reevaluate this study from this point of view. We hope that this study will be helpful for future research and data accumulation in human.

please indicate more details about the preparation of the ovarian cortex; in some studies strips are transplanted instead of biopsies and the number of remaining fibroblasts influences the angiogenesis a lot

Answer: We thank the reviewer for their comments. In this study, we stripped ovarian cortex (figure 1) and stroma of ovarian cortex contains enough amount of fibroblast.

Figure 1. stripping of ovarian tissue

clinical data in human reassume the hormonal activity of the grafts, that are transplanted in a peritoneal pocket in most cases and orientated towards the fallopian tube, after 4 to 8 weeks; therefore the hormone profile would also be beneficial for this manuscript

Answer: We thank the reviewer for their comments. We performed transplantation into the peritoneal pocket, and it is very clinically important to examine E2, FSH, and AMH. In this study, we tried to assess the blood level of E2, FSH, and AMH. However, the maximum amount of whole blood that can be extracted in the laboratory from a mouse (20g) is 1.0-1.5ml and the amount of serum is usually less than 0.5-1ml. Therefore, only one hormone can be measured. In addition, when we compared the results of AMH test between commercial ELIZA kit and blood test of hospital laboratory, we failed to show reliable and consistent results. For these reasons, AMH was measured using western blot from ovarian tissue. We described this point as limitation of our study in discussion section (page 9, line 216-218).

reference 35 is named for the slow-freezing and thawing protocol, but the table in the refrence differs from the design in the manuscript; please check and declare

Answer: We thank the reviewer for their comments. We used modified slow-freezing and thawing protocol. We added the additional reference in material and methods section [13].

especially for the thawing, the exposure time of the tissue in DMSO at room temperature should be as short as possible in order to avoid cellular damage. DMSO and warm temperature harm the tissue; please explain

Answer: We thank the reviewer for their comments. During slow freezing process, all procedures were carried out with ice to prevent tissue damage. In addition, thawed ovarian tissue was washed with phosphate-buffered saline to remove DMSO and we tried to perform transplantation as soon as possible.

Looking forward for the revised version, good luck

Round 2

Reviewer 1 Report

 I still consider that it should be interesting to correlate AMH and primordial follicles, as even if it is produced by antral follicles the one that ovarian tissue cryopreservation should preserve are primordial follicles as they are the effective ovarian reserve.

Author Response

Thank you for kind comment. First of all, you are absolutely right. Hypothetically, primordial follicle density should be positively correlated to AMH. However, we could not show positive results. First, if the blood level of AMH on top of AMH expression in ovarian tissue were assessed, we may have had significant outcomes. Secondly, assessment on the long-term effects on AMH and follicle density after transplantation may have shown significant outcomes. These limitations were described in discussion section of manuscript. Finally, the tendency to increase AMH expression in Z-VAD group was observed in this study (figure 5), although there was no statistical significance between groups. In spite of high follicular density, AMH between groups might not have shown significant difference due to insufficient sample size in addition to other several factors such as ovarian damage caused by freezing-thawing and transplantation process.

Reviewer 2 Report

Dear authors, 

thanks for improving the manuscript and answering all my comments. Please Keep in mind to design future Experiments according to human practice. I support the acceptance now.

Best regards

Author Response

Thank you for kind comment. I will keep in mind your advice.